

# Observations of water vapor within a mid-tropospheric smoke plume using ground-based microwave radiometry

Darren R. Clabo[1]

[1]Program in Atmospheric and Environmental Sciences, South Dakota School of Mines and Technology, Rapid City, 57701,
USA

*Correspondence to*: D. R. Clabo (darren.clabo@sdsmt.edu)

**Abstract.** This study presents an analysis of the water vapor mixing ratio contained within multiple mid-tropospheric smoke plumes as diagnosed by a ground-based passive microwave radiometer. Measurements from the radiometer were compared to smoke opacity as diagnosed from visible satellite imagery on three different days: 12, 16, and 20 August 2013. It was found that the water vapor mixing ratio within the smoke plume could be as much as 20-250% higher than the mixing ratio within the ambient, non-smoke environmental air. Significant intra-smoke plume variability also existed and the mixing ratio was found to be higher (lower) in more optically thick (thin) areas of the plume. This study demonstrates that a radiometer is valuable tool that can be used to remotely measure the water vapor content within smoke plumes.

## 1 Introduction

Wildland fires produce vast amounts of water vapor owing to the moisture released via the combustion reaction (Byram, 1959; Ward, 2001; Parmar et al., 2008). As noted by Ward (2001), for every kilogram of plant material combusted, 0.559 kg of water is released from combustion alone. Additional water vapor is also released prior to ignition during the preheating of the fuel where all moisture contained on or within the fuel must be evaporated before combustion can take place.

Few studies have attempted to directly measure fire-released moisture due to the dynamic and turbulent environment near a wildfire. During the mid-1960s, a collaborative exploratory study was conducted between the United States Department of Defense and the United States Department of Agriculture to examine the effects of urban mass fire behavior (Countryman, 1969). Large fuel beds consisting of pinyon pine-juniper trees were constructed and left to dry until the fuel moisture approximated that of wood in buildings. These fuel beds were then ignited en masse and measurements of $O_2$, $CO_2$, CO, and $H_2O$ were taken. Results (Storey, 1969; Bush, 1969) showed an increase in water vapor concentration over background of approximately ten-fold and twenty-fold for their experiments 460-7 and 760-1, respectively. More recently, Achtemeier (2006) measured the moisture contained within the smoke of fire in the smoldering stage from the Southeastern United States. He found levels of moisture from within the smoke of near negligible amounts to an increase of 39 g kg$^{-1}$ over background levels. Clements et al. (2006) measured the $CO_2$, water vapor, and heat fluxes from a prescribed grass fire near the Texas Gulf Coast



in 2005. Tethersonde measurements revealed an increase in water vapor mixing ratio within the smoke plume of 1.93 g kg$^{-1}$ in the lower layer and in increase of 1.19 g kg$^{-1}$ in the upper layer of the smoke plume. A flux tower was also employed for this experiment and it measured an increase in latent heat flux from 29.7 W m$^{-2}$ to 376.6 W m$^{-2}$. In a similar study, Clements et al. (2007) found increases in water vapor mixing ratio over background levels of ~1 – 2.5 g kg$^{-1}$. In 2008, Parmar et al. reported

on the emission of water vapor from biomass burning finding increases in H$_2$O over background levels of over 7000 ppm (~ 4.4 g kg$^{-1}$) in oak biomass and increases of over 5000 ppm (~ 3.1 g kg$^{-1}$) in both spruce and grass biomasses.

Theoretical experiments using physical arguments and numerical modeling also provide insight into fire-released moisture. Potter (2003) utilized the combustion equation, provided by Ward (2001), and a simple entrainment model to determine reasonable ranges of sensible and latent heat released by a fire. He found increases in water vapor mixing ratio within smoke

plumes can be as little as 0.05 g kg$^{-1}$ and range up to 3 g kg$^{-1}$ above background levels. Potter (2005) illustrated the potential dynamic impacts from fire-released moisture and heat by comparing the plume and associated pyrocumulus lifted condensation level (LCL) and equilibrium level (EL) to nearby radiosonde-derived atmospheric profiles. The EL and LCL from the radiosonde data would only match the plume observations if the idealized surface-based parcel's temperature and water vapor mixing ratio increased by 1 – 3° C and 1 – 3 g kg$^{-1}$, respectively. Luderer et al. (2009) used stoichiometric principles, parcel

theory, and numerical modeling to assess the contributions that wildland fire-produced sensible and latent heat had on pyrocumulus clouds. Through these theoretical considerations they introduced the perturbation ratio: the ratio of the air temperature increase due to sensible heating to the humidity increase due to combustion-released moisture. They found that the perturbation ratio is at least 6.6 K g$^{-1}$ kg but can range to over 30 K g$^{-1}$ kg. This starkly contrasts with the study done by Potter (2005), which infered a perturbation ratio of near unity. Luderer et al. (2009) then used the three-dimensional cloud-

resolving Active Tracer High resolution Atmospheric Model (ATHAM) to diagnose the effect of fire-released moisture on convective columns. From sensitivity studies, they concluded that the effect of the fire-released moisture is small when compared to the effect of the sensible heat released. They also found that the fire-released moisture is a small percentage of the total water budget of a pyrocumulus cloud.

Due to competing results in the present body of literature, additional observational data is needed to understand the amount

and role of fire-released water vapor. The present study uses a passive microwave radiometer (MWR) to measure the water vapor contained within several smoke plumes. Microwave radiometers have the benefit of automatic and continuous retrievals of the water vapor profile (and with some versions, temperature) throughout much of the troposphere. These profiles can then be used in an operational or a quasi-operational state to support fire operations and incident forecasting, and in verification of coupled fire-atmosphere weather prediction models. The hypothesis of this study is twofold: a passive, ground-based

microwave radiometer is suitable for measuring water vapor within biomass smoke, and measureable quantities of water vapor, above ambient levels, are still present within a smoke column many hours after combustion.



## 2 Site and Instrumentation

A Radiometrics (www.radiometrics.com) WVP-1500 microwave radiometer profiler was operational on the campus of the South Dakota School of Mines and Technology (SDSM&T) in Rapid City, South Dakota (44.075, -103.205) during the

5 summer of 2013. This radiometer senses the brightness temperature through 5 channels in the range of 22-30 GHz, on the edge of the 22-GHz water vapor absorption line. The bandwidth for each channel is 300 MHz with an antenna system optical resolution of $4.9 - 6.3°$ (Radiometrics 2006). The WVP-1500 also contains surface sensors that measure the temperature, relative humidity, and barometric pressure at the MWR site. The water vapor density (WVD) vertical resolution of the WVP-1500 is 100 m up through 1 km in altitude AGL with 250 m resolution from $1 - 10$ km AGL. Observations were acquired at

10 10-min intervals.

Numerous studies have shown that MWRs readily produce reliable and accurate water vapor profiles. Elegred et al. (1982), Skoog et al. (1982), and Hogg et al. (1983) demonstrated that dual-channel MWRs can produce integrated precipitable water vapor profiles similar to those of operational radiosondes and with a higher temporal resolution. More recently, multichannel MWRs have come into widespread use (Desrochers, 2005; Padmanabhan et al., 2011; Cimini et al., 2011; Cadeddu et al.,

2013; Ware et al., 2013). These profilers typically use several channels in the 22-30 GHz range, which allow for atmospheric moisture content retrievals within various layers in the atmosphere below 10 km.

The MWR water vapor profile is diagnosed from a neural network (NN; Solheim et al., 1998) technique, supplied by the manufacturer, by utilizing the blackbody temperature measurements within the aforementioned frequencies. Historical Rapid City (KUNR) radiosonde profiles were used to train the NN. The manufacturer states that for the NN technique water vapor

density profiles tend to approach the climatological mean values, especially for heights above 7 km. A TIP calibration procedure (Radiometrics, 2006) was used in order to ensure data quality. Of the available MWR output variables, this study will focus on the path integrated water vapor and the water vapor profiles.

## 3 Methodology

The northern Rocky Mountains were plagued by dozens of wildland fires during August of 2013. Notable fires included the

25 Elk and Pony Complexes, both located east of Boise, Idaho and burning over 131,258 acres and 149,384 acres, respectively; the Miner Paradise Complex located south of Bozeman, Montana which burned over 11,000 acres; and the Druid Complex in northeast Wyoming which also burned over 11,000 acres (http://www.inciweb.nwcg.com). During their life histories, each of these fires produced large smoke columns and several triggered pyrocumulus formation. Furthermore, other wildfires in eastern Montana and northern Idaho were ongoing at the same time, each producing smoke that was able to travel downstream with

30 the ambient environmental winds. The present study leverages a fortuitous data set: several of these large smoke plumes traversed a region that was being interrogated by an MWR.





This study compares visible satellite imagery with the MWR data to diagnose relative water vapor increases within several smoke plumes that passed over western South Dakota in August of 2013. Three criteria were used for specific smoke plumes to be included within this dataset: 1) the smoke plume crosses the radiometer site under cloud free conditions, 2) the smoke plume had a well-defined leading and/or trailing edge, as diagnosed by visible satellite imagery, so that comparisons in the

radiometer data could be made to the ambient (without smoke) atmosphere before and/or after plume passage, and 3) the plume must have at least partially traversed the radiometer site during the daytime so that visible satellite imagery could be used to document the times of plume passage to compare with the MWR. Two cases (16 August 2013 and 20 August 2013) fully met these criteria while one other case (12 August 2013) only partially met the criteria but is included because of unique variations in smoke opacity.

Data at each particular MWR altitude were output as values of water vapor density (g m$^{-3}$). Each value was then converted to mixing ratio ($w$, in g kg$^{-1}$) using the ideal gas law and state variable data from the nearest-in-time KUNR (Rapid City, South Dakota) radiosonde sounding which is launched ~400 m southwest of the MWR site. Pressure was logarithmically interpolated to the corresponding MWR heights, while temperature and mixing ratio data were linearly interpolated. In order to avoid zero-order discontinuities in the figures, only data from one radiosonde was used during the calculations for each of the three case

studies: 1200 UTC 12 August 2013, 0000 UTC 17 August 2013, and 1200 UTC 20 August 2013.

Figure 1 shows the MWR observations of integrated water vapor as compared to the 1200 and 0000 UTC KUNR radiosonde data of the same variable from 8-25 August. There are some notable differences between the radiosonde and MWR observations; however, similar trends in the data are seen. Figures 2-4 compare the calculated $w$ for both the KUNR radiosonde and the MWR for each of the three cases: 19-21 August, 16 August, and 12 August 2013, respectively (the dates are adjusted

to local time). Generally, the MWR agrees well with the data from the radiosonde, albeit with a much smoother profile. Figures 3 and 4 show significant differences between the two instruments below 2 km AGL; however, above this level the $w$ values are in better agreement. These figures demonstrate the relative accuracy of the MWR as compared to the radiosonde for the relevant vertical levels over the duration of the study.

For the satellite and radiometric observation comparisons, it was assumed that a steady-state tropospheric moisture background

was maintained; that is, the water vapor contained within the smoke plume and the variations of the water vapor within were the only causes of any changes in WVD and $w$ throughout the plume passage. The studied smoke plumes were in the mid-troposphere and well-above the planetary boundary layer.

## 4 Results

The results of this study are broken down into two subsections: 1) 20 August 2013 and 2) 12 and 16 August 2013. The 20

August case includes data from the Cloud-Aerosol Lidar and Infrared Pathfinder Satellite Observation (CALIPSO) platform



which adds additional information to support the MWR dataset. The 12 and 16 August cases are presented with only the MWR data and visible satellite imagery but are included in the present paper for completeness.

### 4.1 20 August

During the overnight hours from 19-20 August 2013, a smoke plume traversed the MWR site. Following sunrise, a distinct leading edge was apparent in a dense smoke plume, per GOES-13 visible imagery (Fig. 5a, smoke edge highlighted in yellow). Using photogrammetric techniques based upon steady state progression of the smoke (Figs. 5a-d) as well as KUNR radiosonde wind speed data from the 20 August at 1200Z, it was back-calculated that this optically thick smoke first passed over the radiometer site between 0645 to 0745 UTC, roughly 5 hours prior to sunrise.

The GOES-13 Aerosol and Smoke Product (GASP) was also analyzed in an attempt to determine the aerosol optical depth (AOD) and smoke plume location. Figure 6 is the GASP AOD for the continental United States for 1345 UTC and 1915 UTC (corresponding to Figs. 5a and 5d). There is a concentration of aerosols over the state of South Dakota; however, it appears that the product is not able to fully resolve or correctly diagnose the plume over much of western and central South Dakota. As such, visible satellite imagery was used as the primary means of determining the location of the smoke plume.

At approximately 0903 UTC, the CALIPSO platform passed over western South Dakota, a time at which the dense smoke plume was likely over the radiometer site. The data from CALIPSO instrument can be used to diagnose aerosol height which is critical for proper analysis within the study. From 2-6 km AGL, there is a layer of elevated total attenuated backscatter with values of 1.5-4.5 x $10^{-4}$ km$^{-1}$ sr$^{-1}$ (Fig. 7). This backscatter has been attributed to smoke that was passing over the region (Mike Fromm 2014, personal communication). Additionally, this height also roughly corresponds to a pilot report (PIREP) from over western SD on 20 August that reported a flight level visibility of 5 km at 10,500 ft (3.2 km) AGL, ostensibly due to the presence of smoke at that level (Scott Bachmeier 2013, personal communication).

The smoke continued to travel over the MWR site throughout the day and was optically thick until about 1730 UTC at which point the smoke began to clear, as diagnosed from the visible satellite imagery (Fig. 5c). At approximately 1831 UTC, deep moist convection (Fig. 5d) initiated over the northern Black Hills northwest of Rapid City. As the convection continued to build, cumulonimbus incus developed, with the edge of the anvil eventually passing over Rapid City at 1945 UTC.

To supplement the GOES-13 data, the MODIS imagery from both the Terra and Aqua platforms were also analyzed. Smoke can clearly be seen over/near the radiometer location in the true color imagery (Fig. 8a, b). MODIS AOD imagery was then examined (Fig. 9a, b) by utilizing the Giovanni Infrastructure (Acker and Leptouch, 2007). Enhanced AOD at 550 nm was seen over the radiometer site in both Fig. 9a and 9b although the AOD does appear to diminish between the Terra and Aqua passes at ~1830 UTC and ~1940 UTC, respectively.

Figure 10 is a time series plot of the radiometric observations of $w$ at 250 m increments from 3-6 km (the primary smoke layer as diagnosed from the PIREP and CALIPSO observations) from 2300 UTC on 19 August to 0550 UTC on 21 August.



Considering the 3-6 km average, $w$ is higher from the time period of 0645-1550 UTC than it is either preceding or immediately following that period. This corresponds to the time of which smoke was noted in the air above Rapid City, as noted by the GEOS-13 data. The large increase in $w$ after 1830 corresponds very well with the time in which the cirrus anvil from the nearby thunderstorm encroaches on Rapid City.

Comparing $w$ values during the smoke plume passage with $w$ values in the pre- and post-smoke plume timeframe, it is shown that on average 24% more water vapor is contained in the 3- to 6-km layer. The 3- to 6-km average $w$ for the hour prior to smoke passage was approximately 1.24 g kg$^{-1}$ while the $w$ rose to an average of 1.66 g kg$^{-1}$ during plume passage with a peak value of 2.20 g kg$^{-1}$. During the hour following the smoke passage and before the cirrus anvil arrival, there is an average of 7% less water vapor in the 3- to 6-km layer, with a minimum 3- to 6-km average $w$ of 1.39 g kg$^{-1}$. The elevated levels of $w$
correspond well with the time of smoke plume passage.

**4.2 16 and 12 August**

Figures 11a-d are GOES-13 imagery showing wildland fire smoke traversing South Dakota on 16 August 2013. The yellow line highlights the leading edge of the smoke while the yellow arrow shows the MWR site. The leading edge of the smoke plume impinges on the MWR site at approximately 2130 (Fig. 11b). The smoke then slowly progresses eastward in the weak
upper level flow. Imagery from the MODIS instrument onboard the Aqua satellite also shows the smoke (Fig. 12) nearing the MWR site at approximately 2100 UTC. As the sun begins to set, the smoke plume becomes more apparent due to a longer light path length and by 0025 UTC on 17 August, a vast pool of smoke has covered the western quarter of South Dakota. Additionally, at sunrise on 17 August smoke can still be seen via visible satellite imagery (not shown) and it is assumed that smoke was present above the radiometer site throughout the night of 16-17 August. For both the 16 August and 12 August
(below) case, both the GASP and MODIS AOD data were analyzed but are not included in the analysis for brevity.

Figure 13 is a time series plot of $w$ from 16 August, similar to that of Fig. 10. Although no CALIPSO data or PIREPS were available for this case, observations of $w$ are made from 3- to 6-km AGL assuming that the smoke plume was once again contained primarily within this layer. In large part, the average $w$ in the 3- to 6-km layer decreases from 1400-2140 UTC. During this period the smoke plume is still well off to the west and a weak cumulus field is dissipating to the southeast of the
area (Figs. 11a-d). The $w$ tended to increase after this period through 0700 UTC corresponding to the period of time when smoke was observed moving over the radiometer side. Following 0700 UTC the rate of increase in $w$ tended to diminish.

Averaging the 3- to 6-km $w$ from the time period from 2100-2200 UTC in Fig. 13 reveals a value of 1.46 g kg$^{-1}$. From 2200 UTC through 0700 UTC the average $w$ within the layer rises by over 3.0 g kg$^{-1}$ to a 3- to 6-km layer average $w$ of 3.77 g kg$^{-1}$ representing an increase of over 250%. An examination of both the visible satellite imagery (during the daylight) and of the
infrared imagery (at night) reveals no cloud cover during this time. This increase in $w$ is then likely due to the moisture associated with the advecting/thickening smoke and/or to moisture advection within the layer from an unknown source.





A smoke plume was evident on GOES-13 visible imagery over South Dakota, and much of the northern Great Plains region, throughout the day on 12 August (Fig. 14a-d). The smoke is also noticeable on both Terra and Aqua polar-orbiting satellite passes (Fig. 15a-b). The smoke plume on this day was fairly diffuse and without a distinct leading or trailing edge. This made an analysis of changes in atmospheric moisture content due to the smoke plume, as compared to the without-smoke environment all but impossible. However, as the satellite imagery shows, there is variation in smoke plume density over the MWR site. The focus of this discussion will be on these subtle differences.

A loop of the visible satellite imagery from 12 August (available from http://www.mmm.ucar.edu/imagearchive/) shows how the smoke appears to dissipate over the MWR site. The sun angle undoubtable affects the appearance or optical thickness of the smoke in the satellite imagery: as the sun angle increases, the path length of the light within the smoke decreases, further decreasing the light reflected back to the satellite. The smoke is optically thickest at 1345 UTC (Fig. 14a) and a decrease in thickness is observed through 1645 UTC (Fig. 14b). From 1645 UTC through roughly 1945 UTC (corresponding to Figs. 14b and 14c, respectively), the opacity of the smoke appears fairly uniform over the MWR site. Development of cumulus clouds begins over the northern Black Hills with initiation starting near 1900 UTC. By 2145 UTC, the radiometer site has been covered by the anvil of a cumulonimbus (Fig. 14d).

Similar to Figs. 10 and 13, Fig. 16 is a time series plot of $w$ from 12 August. Paralleling the previous two cases, $w$ was approximated in the 3- to 6-km layer. A gradual increase in $w$ is noted from 0630-1330 UTC after which the $w$ decreases through 1600 UTC. As seen in the satellite imagery, the smoke density appears to decrease after 1345 UTC. The density then remain constant from 1600 through 2000 UTC. From 2000-2400 UTC $w$ increases, corresponding to the time when the cumulonimbus anvil was over the radiometer site. The 3- to 6-km layer average $w$ from 1320 to 1420 UTC (roughly corresponding to the 1345 UTC satellite image and peak values of $w$) was calculated to be 1.79 g kg$^{-1}$. From this maximum, the $w$ decreases to a 1600-2000 UTC time-average 3- to 6-km layer average of 1.27 g m$^{-3}$. This represents a decrease in $w$ of nearly 30%.

These relative increases and decreases in $w$ match nicely with the satellite observations presented previously: it appears that as the opacity of the smoke increases, the water vapor density signal as well as the $w$ from the MWR increase as well. This implies that enhanced levels of smoke, as diagnosed from the visible satellite imagery, may contain higher amounts of water vapor.

## 5 Discussion

This study covers three case studies and some intriguing results were obtained. It was shown that within smoke plumes, $w$ increased from 20% to 250% over background levels. This is obviously a large range of variability. Regardless, one similarity does exist between the cases: during times of increased smoke opaqueness, there were elevated levels of $w$.



The results presented here mirror those of past studies which, as outlined in the background section, found increases in water vapor content within smoke plumes. The results of Storey (1969) and Bush (1969), measuring smoke emanating from large masses of pinyon pine, show increases in water vapor over background of 1000-2000%. Clements et al. (2006, 2007) showed net moisture increases within a smoke plume emanating from coastal grasses of 20-30% above background levels. Achtemeier

(2006) measuring the mixing ratio of smoke directly above a smoldering fire source found increases of 0-400% above background levels. However, these observations were all made within line of sight of the fire and only within a few to tens of meters above the ground. This contrasts with the present study that indirectly measured $w$ from within smoke plumes emanating from fires on the order of $10^3$ km away and several $10^3$ m above the ground. Admittedly, the comparisons between the listed past studies and the present can only go so far; however, the similarities in the water vapor increases are striking.

The obvious next step is to attempt to understand why there was an increase in the $w$ during the time of the smoke passage. As noted in the introduction, considerable water vapor is released during combustion due to the chemical reaction itself and to the evaporation of moisture contained within the fuels. It is easy to draw the conclusion that the increase in $w$ noted in the MWR observations was due to the release of water vapor from the fire and the water vapor advected downstream in sync with the smoke plume. However, this is may not be the case.

During the production of the smoke events presented here, pyrocumulus were present above some, if not all, of the fires that produced the smoke plumes. Pyrocumulus are cumulus clouds that ostensibly obtain some of their energy, via sensible and latent heat release, from the fire itself. What is not known is what proportion of the energy is derived from the fire or from the background atmosphere. Luderer et al. (2009) presented one case study for the Chisolm Fire and found that the fire-released moisture only accounted from ~10% of the total water budget within the plume with the remainder being entrained from the

ambient atmosphere. If this theory holds for the present study, much of the increase in water vapor seen within the smoke plumes may have been a result of boundary layer moisture, unrelated to the wildland fire, which was entrained into the plume and the pyrocumulus cloud, lofted into the mid-troposphere, and transported to the skies above western South Dakota. Although this idea is merely a conjecture at the present time, it may be a viable direction for further analyses of these cases.

The purpose of this paper is not to speculate on the direct source of moisture, but to present measurements of moisture within

smoke plumes by using a novel technique. Applying this technique closer to the fire itself may prove useful in determining the relative amount of moisture within the smoke plume as compared to the ambient air. Furthermore, observations of smoke plumes known not to have been influenced by the effects of pyrocumulus may prove most beneficial as the entrainment of boundary layer moisture could potentially be ruled out. Data of these types may be helpful in validating and/or improving classic numerical as well as coupled fire-atmosphere weather prediction models. Real-time radiometric data may also be useful

in diagnosing the potential for pyrocumulus formation and resulting impacts to the on-ground fire behavior.



## 6 Conclusions

A microwave radiometer was used to measure atmospheric moisture present before, during, and after the passage of transient smoke plumes caused by wildland fires. Smoke plumes and their borders were readily identified in both visible satellite imagery from GOES-13 as well as from MODIS true-color imagery on the Terra and Aqua satellites. Results from the study

show that a radiometer is able to measure increased levels of water vapor within biomass smoke plumes.

Increases in mixing ratio within the plume were found to be between 20% and 250% above ambient environmental levels, assumed for this study to be outside the visible edge of the plume itself. Additionally, variations in smoke optical thickness, as diagnosed from visible satellite imagery, were found to correspond to variations in mixing ratio: optically thicker smoke produced higher mixing ratios. This paper assumes that it is the water vapor originally generated by combustion and inherently

contained within the visible aerosols of the smoke that was detected by the radiometer. However, some of the water vapor within the smoke plumes may have been vapor entrained into pyrocumulus clouds from a non-fire affected boundary layer, which was then lofted and advected to the studied region.

A passive microwave radiometer may prove to be a valuable tool in diagnosing the amount of water vapor released during combustion in a wildland fire. The inexpensive nature of the instrument, combined with its automation and ease of use makes

the radiometer a tool of particular interest in measuring water vapor levels above and downstream of a fire. Potential future uses for this information include inputs to numerical prediction systems, validation of fire-atmospheric models, and real-time forecasting operations.

### Acknowledgements

The author would like to acknowledge Drs. Andy Detwiler, Paul Smith, and Mike Fromm for their helpful edits and

commentary. Support for this project was made possible by the State of South Dakota. Analyses and visualizations used in this paper were produced in part with the Giovanni online data system, developed and maintained by the NASA GES DISC.

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



**FIGURES**

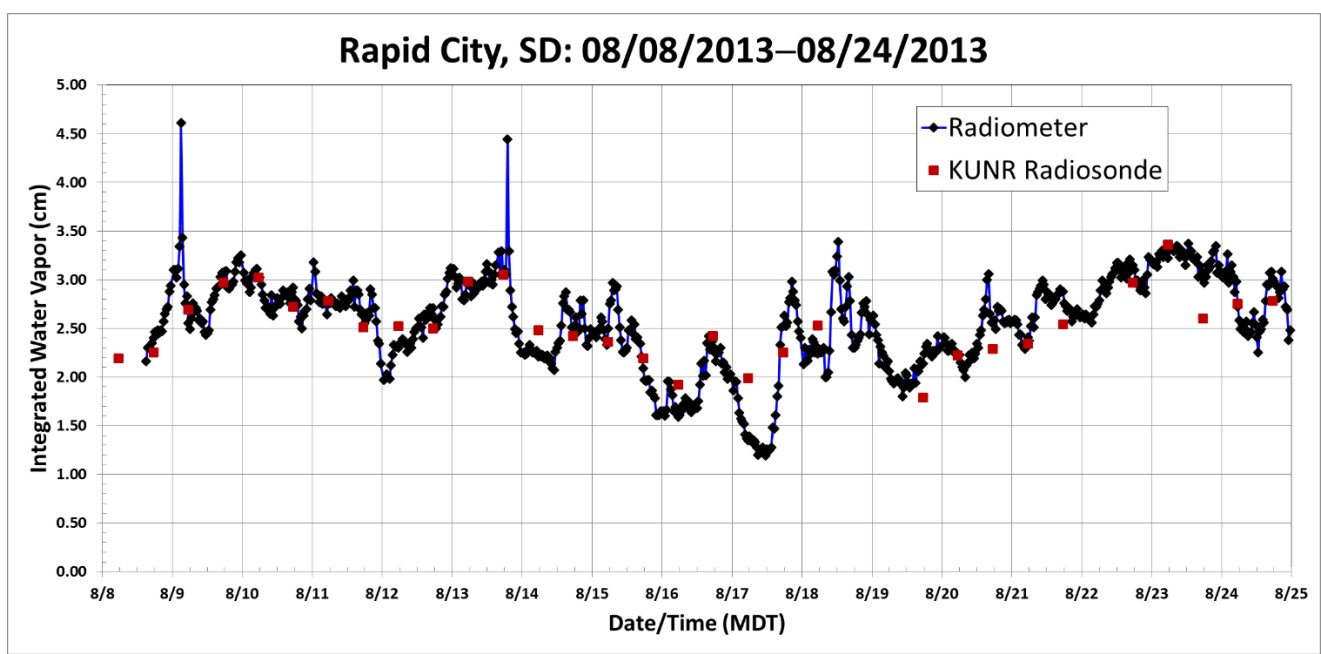

Figure 1. Time series plot from 8 – 25 August 2013 of integrated water vapor (cm) for both the microwave radiometer and

5   for the KUNR radiosonde. Figure courtesy of Matthew Bunkers, National Weather Service Forecast Office Rapid City,

South Dakota, USA.





Figure 2. A comparison of mixing ratio (g kg$^{-1}$) between the MWR and the KUNR radiosonde for 1200 UTC on 20 August
2013.





Figure 3. As in Fig. 2 except for 0000 UTC on 17 August 2013.



Figure 4. As ii Figs. 2 and 3 except for 1200 UTC on 12 August 2013.







Figure 5. GOES-13 visible satellite imagery for the northern Great Plains of the United States for 20 August 2013 at a) 1345 UTC, b) 1500 UTC, c) 1730 UTC, and d) 1915 UTC. White outlines show US states while the yellow arrow points to the radiometer site. The yellow line depicts the easternmost edge of the dense visible smoke plume which is traveling east with time. Images taken from www.aviationweather.gov.




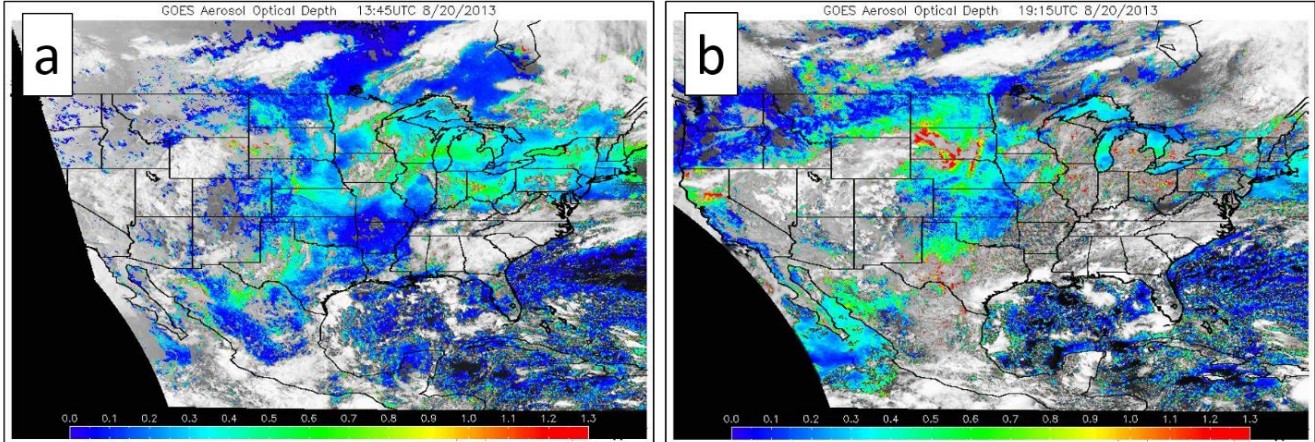

Figure 6. GOES-13 Aerosol and Smoke Product (GASP) imagery for 20 August 2013 at a) 1345 UTC and b) 1915 UTC.





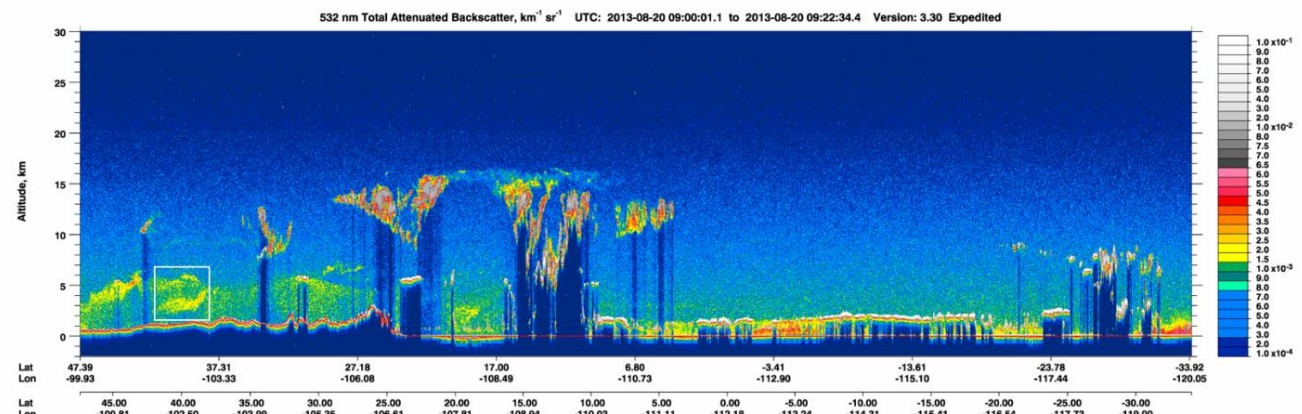

Figure 7. CALIPSO lidar-derived total attenuated backscatter imagery from 20 August 2013. This satellite passed over western SD (~43° N, 103° W) at approximately 0903 UTC. Enhanced backscatter from the smoke particles can be seen in the 2- to 6-km layer with values ranging from $1.5 - 4.5 \times 10^{-4}$ km$^{-1}$ sr$^{-1}$ (highlighted by the white box near the left side of the figure).



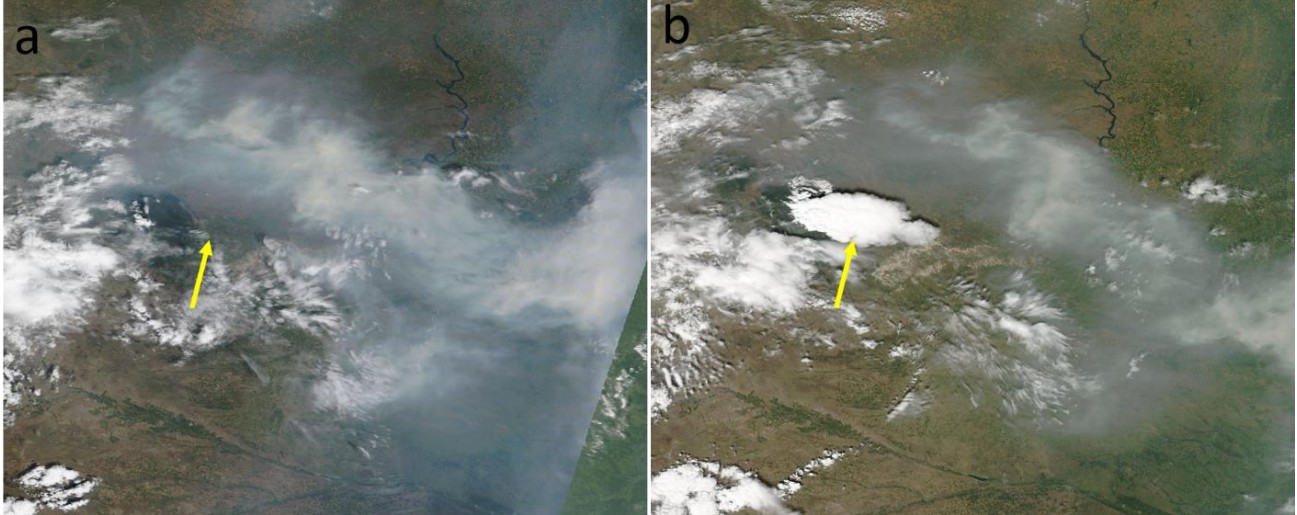

Figure 8. MODIS imagery taken on 20 August 2013 from the a) Terra (~1830 UTC) and b) Aqua (~1940 UTC) satellites.
The yellow arrow points to the radiometer location.





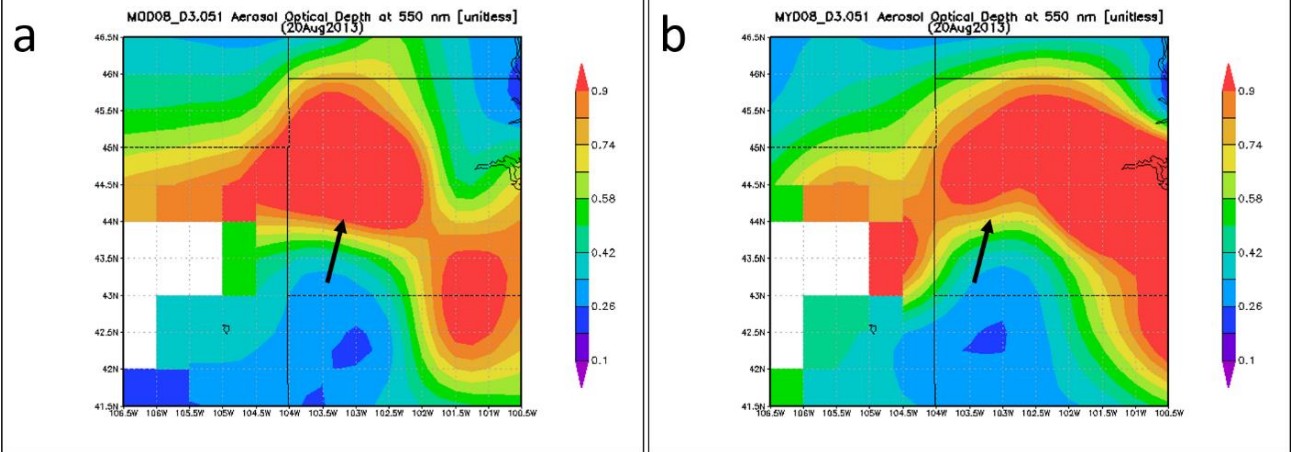

Figure 9. MODIS AOD data from 20 August 2013 from the a) Terra (~1830 UTC) and b) Aqua (~1940 UTC) platforms. The black arrow denotes the location of the MWR.




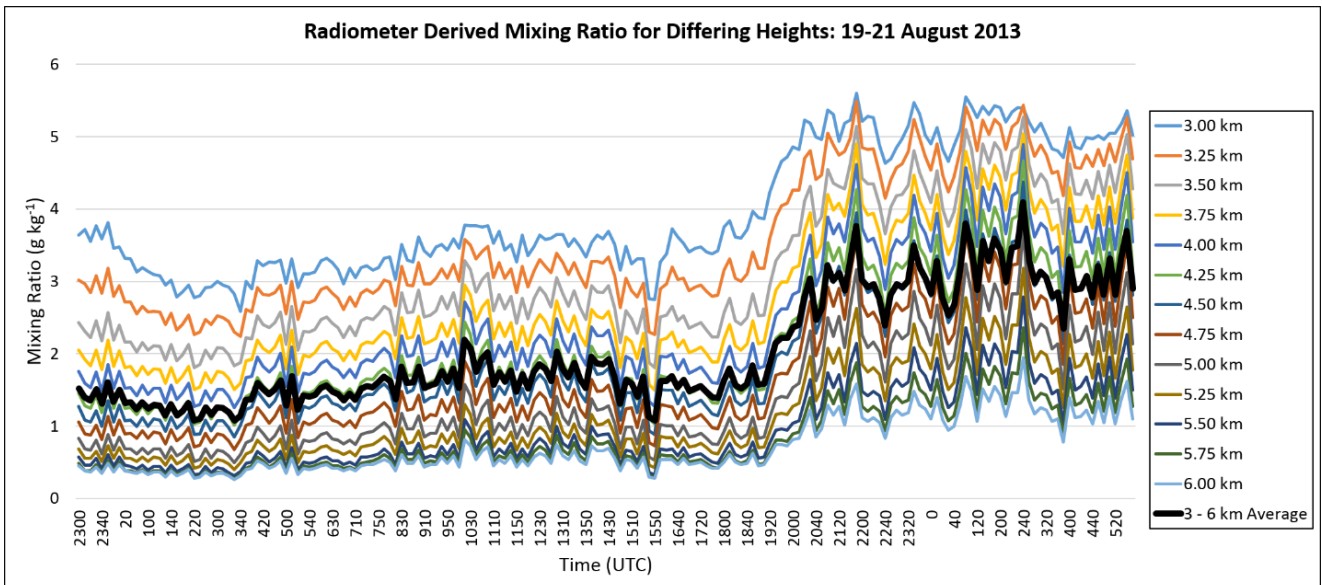

Figure 10. Time series plot of microwave radiometer derived mixing ratio (g kg$^{-1}$) at various levels for 19-21 August 2013. The thick black line represents the average mixing ratio in the 3- to 6-km layer.



Figure 11. As in Fig. 5 but for 16 August 2013 at a) 2015 UTC, b) 2130 UTC, c) 2245 UTC, and d) 17 August 2013 at 0025 UTC.





Figure 12. MODIS Aqua imagery of the smoke plume on 16 August 2013 at ~2045 UTC. The yellow line outlines the easternmost edge of the visible smoke plume while the yellow arrow points to the radiometer site.





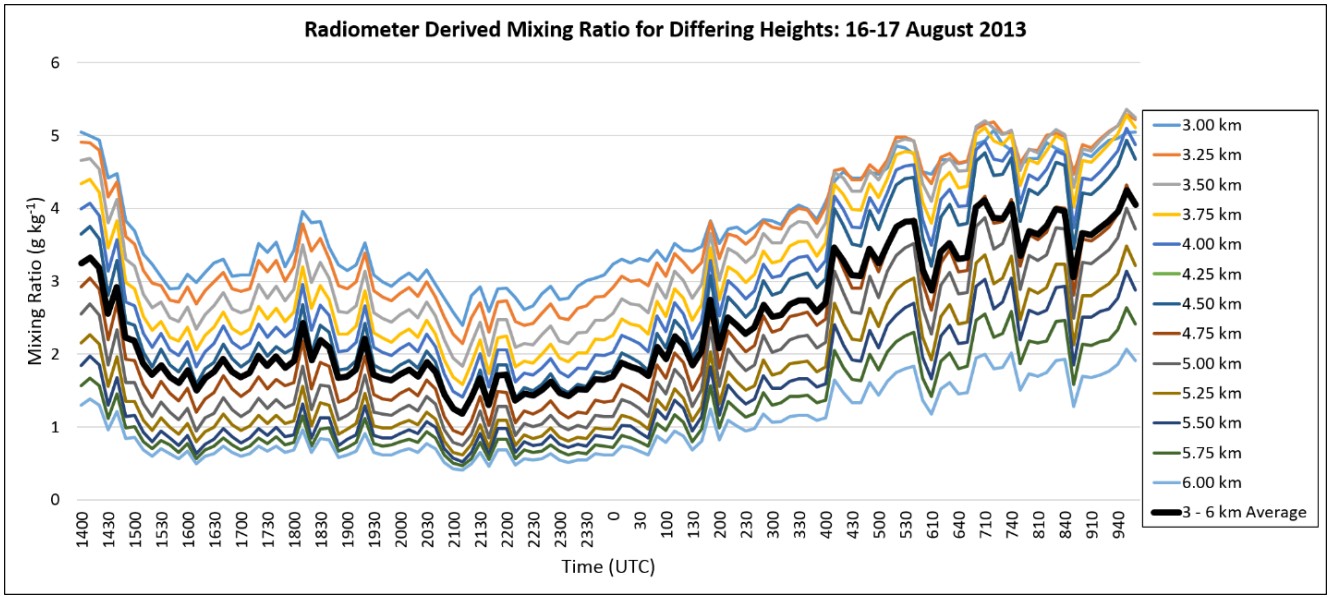

Figure 13. As in Fig. 10 but for 16 – 17 August 2013.





Figure 14. As in Figs. 5 and 11 but for 12 August 2013 at a) 1345 UTC, b) 1345 UTC, c) 1945 UTC, and d) 2145 UTC.
There is no smoke edge highlighted in these images because the plume was quite diffuse and lacked distinguishing
boundaries. Images taken from www.aviationweather.gov.




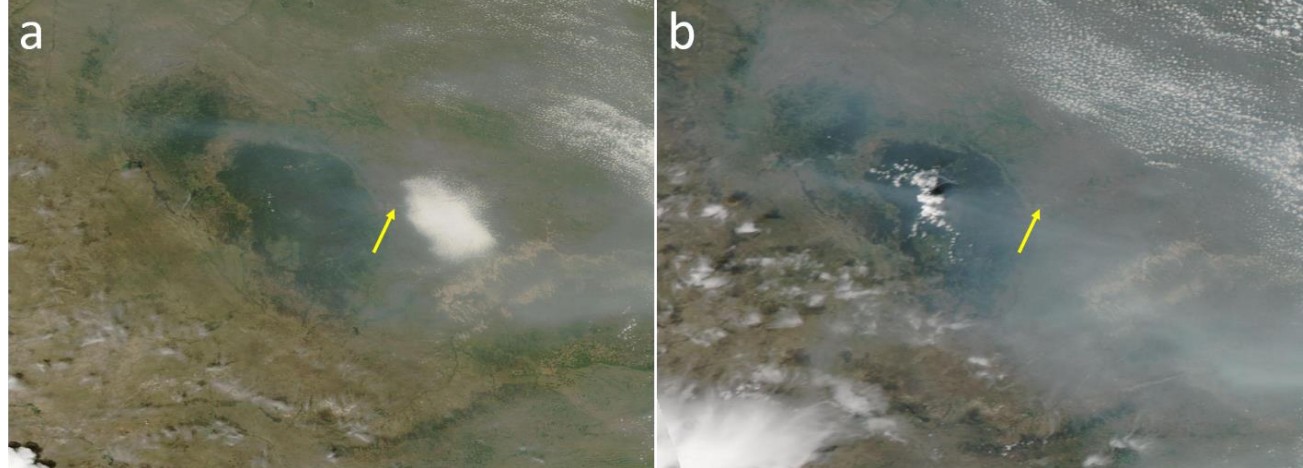

Figure 15. MODIS imagery taken on 12 August 2013 from the a) Terra (~1745 UTC) and b) Aqua (~1915 UTC) satellites.
The yellow arrow points to the radiometer location.





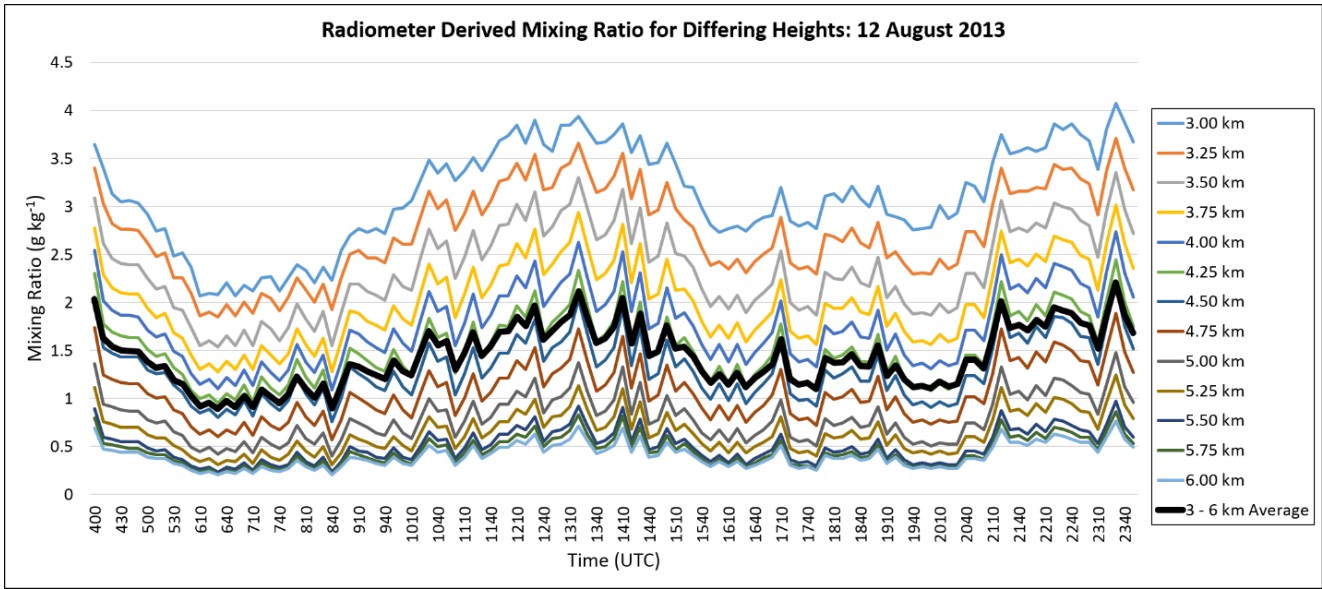

Figure 16. As in Figs. 10 and 13 but for 12 August 2013.