# Peer review of "Observations of water vapor within a mid-tropospheric smoke plume using ground-based microwave radiometry"

_Atmospheric Measurement Techniques, 2016_

## Referee Comment (RC1) · Anonymous Referee #2 · 11 May 2016

**Review for paper „ Observations of water vapor within a mid-tropospheric smoke plume using ground-based microwave radiometry"**

**by D. R. Clabo**
**submitted to Atmospheric Measurement Techniques**

**General vote: Major revision**

**General comments:**

The manuscript presents measurements from a microwave radiometer in the vicinity of forest fires in order to detect water vapor increases due to combustion reactions. The approach is interesting, however the number of cases and the lack of a statistical analysis makes the hypothesis that signals in the measurements are related to the smoke plume fairly vague.

I consider the paper to be published only after some major changes, such as:

- Checking whether the increase in water vapor on the case study days is significantly higher than on other days. A good way for that would be also checking the diurnal variation of PWV and WV profiles or typical diurnal cycles of water vapor and compare smoke-free with smoke days.

- Using backward trajectories to verify if the airmass considered stems in fact from the smoke region. This would also allow to answer the speculation of the smoke plume height for 2 of 3 case studies. You could use HYSPLIT for that.

- State more clearly that the vertical resolution of water vapor profiles by microwave radiometers is very coarse and that only 2 independent layers can be derived over the whole profile. Especially inversions in mid and high troposphere cannot be determined by MWR. Using the integrated water vapor might give more robust results, as the PWV retrieval has much less uncertainties than WV profiles retrieved from MWR.

- What did you do in case of rainy periods? Did you filter out these data? The measurements are not meaningful at all during rain due to scattering by (large) raindrops. Please mention that in section 2 or 3. See also Figure 1 for that!

- Consider to reduce the amount of figures, as some information is redundant.

**Specific comments, Technical corrections:**

- page 2, line 10: What causes the differences between 0.05 and 3 g kg$^{-1}$ ? Please specify what causes this differences!

- page 2, line 22: What is "small percentage"? Do you have a number for that? Check if there is a number in the reference paper!

- page 3, line 9: The effective resolution is much coarser. There are only 2 independent vertical layers which can be detected. For more details check e.g. Caddedu, 2013 or Löhnert and Maier, 2012 or Güldner and Spänkuch, 2001. Note that the weighting functions for WV profiles do strongly overlap. Please state the uncertainty level for the profiles!

- page 3, line 12-13: "integrated precipitable water vapor profiles" > this expression is nonsense.
  For the integrated value write either "integrated water vapor" or "precipitable water". "water vapor profiles" would correspond to vertically resolved retrievals.

- page 3, line 16: Please keep in mind that the vertical resolution is very coarse and that only 2 independent layers can be detected. The vertical information comes only due to the pressure broadening of the water vapor line, this signal is relatively small. Water vapor profiles from MWR are generally unable to capture inversions.

- page 3, line 18: write "brightness temperature" instead of "blackbody temperature"

- page 4, line 30: reference for CALIPSO! e.g. Winker et al., 2010 or Omar et al., 2009 for products

- page 5, line 16-18 / Figure 7: Did you look into the aerosol classification product from CALIPSO? This is also available online!

- page 5, line 19: SD means South Dakota?

- page 5, line 22: better "approximately 1830". 1831 is too specific to be approximately...

- page 6, line 9: is that result significant? 7 % less water vapor with an uncertainty which is in my knowledge much higher than that

- page 6, line 28: what is the increase in PWV (integrated water vapor)? is there any large scale water vapor transport? check trajectories and weather charts for that!

- page 7, line 29: how do you know background levels? the increase of 250 % might also be caused by synoptic scale water vapor advection

- page 8, lines 20-23: for that, a thorough check of trajectories could give an answer

- page 8, line 25: MWR water vapor profiles are not really a "novel technique", it's just the first time to derive changes in

**Comments to figures:**

- Figure 1: What are the outliers on 8/9 and 8/13? Are the data filtered for rain?

- Figures 2, 3, and 4 could be put into one plot (maybe with subplots).

- Fig. 7: mention figure reference (CALIPSO website)

- In my opinion, it would be enough to show either 1 line per km (3,4,5,6 km) or PWV in the time series plots (Fig. 10, 13, 16). There is no additional information in the other lines!

- Fig. 11 does not show so much – you could skip that since you have Fig. 12 for the same day

---

## Referee Comment (RC2) · Anonymous Referee #1 · 11 May 2016

This manuscript presents several case studies of smoke plumes observed by a microwave radiometer, also using visible satellite imagery to identify the plume locations. The study is interesting and potentially useful, but conclusions on the presence of elevated moisture in smoke plumes are not fully convincing and require further analysis.

Major comments

1) The comparison between moisture values inside and outside the plume needs to be done with more statistical and scientific rigour. There are two tests that need to be passed:

(i) Statistical significance. This should be relatively easy to determine using a Student's

t-test on the hypothesis that mean moisture values are higher inside than outside the plume. In general it would be helpful to see some tables giving the exact time periods considered 'inside' and 'outside' the plume as this is not always easy to glean from the text. Also the tables could contain the mean, standard deviation, and number of samples in each period, from which it will be possible to compute the statistical significance of the difference in water vapour between inside and outside the plume.

(ii) Physical significance. As illustrated by the various timeseries of water vapour from the radiometer, there is a lot of background temporal variability in WV as different airmasses are advected over the observation site. The task the authors face (possibly difficult) is to show that the plume moisture values have been elevated above and beyond this natural variability. In the absence of any smoke plumes or cloud, on days with similar weather conditions to those in the case studies, if we were to pick a number of 2-hour periods at random and compute the difference in WV between the first and second hour in that time period, what size WV difference could be expected? The in-plume WV elevation has to fall outside the PDF of this background variability to be significant.

2) In general it would be good to see more information on the quality and characteristics of the radiometer observations. In particular one of the conclusions of the study is that a radiometer is useful for evaluating elevated moisture levels in plumes. To support this conclusion, it would be good to evaluate the error in the water vapour retrieval by comparison to the nearby radiosonde ascents (i.e. to give the mean and standard deviations of typical difference between the radiometer retrieval and the sonde). Since the authors are examining 3-6km average mixing ratio, it would be most useful to know the error characteristics of this average.

Minor commments

1) Section 2, on the radiometer: Although the WV retrievals are performed on a 0.25km grid in the vertical, as the comparisons to radisonde profiles illustrate, the true vertical

resolution is likely to be much lower. It would be useful to give this true resolution (noting that a neural network retrieval cannot supply this information, but there must have been studies using physical inversion techniques applied to similar radiometers that can supply this information).

2) Is anything known about the radiative impact of smoke aerosol at frequencies used by the microwave radiometer? Presumably it is minimal, but it would be good to see some physical confirmation of this.

3) Figure 6: Some explanation of the meaning and units of the colour scale needs to be given here. In particular the significance of the grey areas is not clear.

4) Figure 11: It is impossible to distinguish the aerosol zone from the ambient air, especially in panels b-d. Some adjustments may need to be made (e.g. to the colour scale?) on these figures.

5) Figure 14, caption: Are panels (a) and (b) really both 1345 UTC?

---

## Author Comment (AC1) · 15 Aug 2016

Review for paper „ Observations of water vapor within a mid-tropospheric smoke plume using ground-based microwave radiometry"

by D. R. Clabo

submitted to Atmospheric Measurement Techniques

General vote: Major revision

General comments:

The manuscript presents measurements from a microwave radiometer in the vicinity of forest fires in order to detect water vapor increases due to combustion reactions. The approach is interesting, however the number of cases and the lack of a statistical analysis makes the hypothesis that signals in the measurements are related to the smoke plume fairly vague.

**Thank you for your comments and I appreciate the time you've spent to help improve this manuscript. My responses to your comments are given in bold directly below the individual bullet points.**

I consider the paper to be published only after some major changes, such as:

- Checking whether the increase in water vapor on the case study days is significantly higher than on other days. A good way for that would be also checking the diurnal variation of PWV and WV profiles or typical diurnal cycles of water vapor and compare smoke-free with smoke days.

> **I did not compare the case studies to other smoke-free days as we were somewhat limited in the number of overall radiometer observations. Additionally, the intra-day variation could be quite substantial and is out of the primary scope of the present paper. However, knowledge of the day-to-day variation in water vapor in that layer would make for an interesting study on its own.**

- Using backward trajectories to verify if the airmass considered stems in fact from the smoke region. This would also allow to answer the speculation of the smoke plume height for 2 of 3 case studies. You could use HYSPLIT for that.

> **I did utilize HYSPLIT in an attempt to determine smoke plume height over the radiometer. However, the HYSPLIT trajectories are very sensitive to plume injection height over the smoke source and those heights are unknown. I could not verify an injection height for the smoke therefore was not confident in the smoke height provided by the trajectory analysis.**

- State more clearly that the vertical resolution of water vapor profiles by microwave radiometers is very coarse and that only 2 independent layers can be derived over the whole profile. Especially inversions in mid and high troposphere cannot be determined by MWR. Using the integrated water vapor might give more robust results, as the PWV retrieval has much less uncertainties than WV profiles retrieved from MWR.

> **This has been addressed in Section 2 with the inclusion of additional references. The integrated water vapor was not used because it would be heavily weighted towards the lowest layer of the atmosphere where most of the water vapor is contained.**

- What did you do in case of rainy periods? Did you filter out these data? The measurements are not meaningful at all during rain due to scattering by (large) raindrops. Please mention that in section 2 or 3. See also Figure 1 for that!

**This has been addressed in Section 3. Additionally, there was no rain observed during the periods of study.**

- Consider to reduce the amount of figures, as some information is redundant.

**I did remove MODIS imagery for two of the case studies to lessen the number of figures.**

Specific comments, Technical corrections:

- page 2, line 10: What causes the differences between 0.05 and 3 g kg-1 ? Please specify what causes this differences!

**This is outside the scope of the present paper. The referenced paper describes the involved calculations that gave the differences.**

- page 2, line 22: What is "small percentage"? Do you have a number for that? Check if there is a number in the reference paper!

**This has been addressed.**

- page 3, line 9: The effective resolution is much coarser. There are only 2 independent vertical layers which can be detected. For more details check e.g. Caddedu, 2013 or Löhnert and Maier, 2012 or Güldner and Spänkuch, 2001. Note that the weighting functions for WV profiles do strongly overlap. Please state the uncertainty level for the profiles!

**This has been addressed with the addition of several more references in Section 2.**

- page 3, line 12-13: "integrated precipitable water vapor profiles" > this expression is nonsense. For the integrated value write either "integrated water vapor" or "precipitable water". "water vapor profiles" would correspond to vertically resolved retrievals.

**This has been addressed.**

- page 3, line 16: Please keep in mind that the vertical resolution is very coarse and that only 2 independent layers can be detected. The vertical information comes only due to the pressure broadening of the water vapor line, this signal is relatively small. Water vapor profiles from MWR are generally unable to capture inversions.

**This has been addressed.**

- page 3, line 18: write "brightness temperature" instead of "blackbody temperature"

**This has been addressed.**

- page 4, line 30: reference for CALIPSO! e.g. Winker et al., 2010 or Omar et al., 2009 for products

**This has been addressed.**

- page 5, line 16-18 / Figure 7: Did you look into the aerosol classification product from CALIPSO? This is also available online!

   **This has been addressed.**

- page 5, line 19: SD means South Dakota?

   **This has been addressed.**

- page 5, line 22: better "approximately 1830". 1831 is too specific to be approximately…

   **This has been addressed.**

- page 6, line 9: is that result significant? 7 % less water vapor with an uncertainty which is in my knowledge much higher than that

   **This has been addressed with the inclusion of a Student's T Test.**

- page 6, line 28: what is the increase in PWV (integrated water vapor)? is there any large scale water vapor transport? check trajectories and weather charts for that!

   **This is an important point. However, I did include wording that it was assumed there was no large scale water vapor transport during the periods of study. Spatial resolution on the "weather charts" is insufficient to determine transport of moisture at the levels I am studying.**

- page 7, line 29: how do you know background levels? the increase of 250 % might also be caused by synoptic scale water vapor advection

   **Admittedly, it could be. I am assuming that it wasn't.**

- page 8, lines 20-23: for that, a thorough check of trajectories could give an answer

   **See above responses.**

- page 8, line 25: MWR water vapor profiles are not really a "novel technique", it's just the first time to derive changes in

   **This has been addressed.**

Comments to figures:

- Figure 1: What are the outliers on 8/9 and 8/13? Are the data filtered for rain?

   **This has been addressed.**

- Figures 2, 3, and 4 could be put into one plot (maybe with subplots).

   **This is an interesting idea but I did not change the plots. There is no impact to the final paper.**

- Fig. 7: mention figure reference (CALIPSO website)

   **This has been addressed.**

- In my opinion, it would be enough to show either 1 line per km (3,4,5,6 km) or PWV in the time series plots (Fig. 10, 13, 16). There is no additional information in the other lines!

**As noted, the vertical resolution is very course above the boundary layer. I do think that the inclusion of all of the lines in the figures between 3-6 km does illustrate that each line may not provide additional information. I think that is an important piece of the figures.**

- Fig. 11 does not show so much – you could skip that since you have Fig. 12 for the same day

**Figure 12 was removed and Figure 11 was enhanced to better show the smoke plume.**

---

## Author Comment (AC2) · 15 Aug 2016

This manuscript presents several case studies of smoke plumes observed by a microwave radiometer, also using visible satellite imagery to identify the plume locations. The study is interesting and potentially useful, but conclusions on the presence of elevated moisture in smoke plumes are not fully convincing and require further analysis.

**Thank you for your comments and I appreciate the time you've taken to review this paper. I have addressed each point you've made and my responses to each point are given in bold text below the individual comments. I primarily focused on improving the statistic rigor of the paper.**

Major comments

1) The comparison between moisture values inside and outside the plume needs to be done with more statistical and scientific rigour. There are two tests that need to be passed:

(i) Statistical significance. This should be relatively easy to determine using a Student's t-test on the hypothesis that mean moisture values are higher inside than outside the plume. In general it would be helpful to see some tables giving the exact time periods considered 'inside' and 'outside' the plume as this is not always easy to glean from the text. Also the tables could contain the mean, standard deviation, and number of samples in each period, from which it will be possible to compute the statistical significance of the difference in water vapour between inside and outside the plume.

**Thank you for making this comment as I believe the paper is now in much better shape. I have added the results of a Student's t-test for the 20 August case.**

(ii) Physical significance. As illustrated by the various timeseries of water vapour from the radiometer, there is a lot of background temporal variability in WV as different airmasses are advected over the observation site. The task the authors face (possibly difficult) is to show that the plume moisture values have been elevated above and beyond this natural variability. In the absence of any smoke plumes or cloud, on days with similar weather conditions to those in the case studies, if we were to pick a number of 2-hour periods at random and compute the difference in WV between the first and second hour in that time period, what size WV difference could be expected? The in-plume WV elevation has to fall outside the PDF of this background variability to be significant.

**This is a very interesting point and this study does beg the question of "what is the natural variability of water vapor over western South Dakota." However, I do believe that answering this question is outside the scope of the present paper and may likely be worthy of a study on its own. I do not disagree that the issue of physical significance is important but a thorough examination of the natural water vapor variability would not be possible with the limited radiometer data collected.**

2) In general it would be good to see more information on the quality and characteristics of the radiometer observations. In particular one of the conclusions of the study is that a radiometer is useful for evaluating elevated moisture levels in plumes. To support this conclusion, it would be good to evaluate the error in the water vapour retrieval by comparison to the nearby radiosonde ascents (i.e. to give the mean and standard deviations of typical difference between the radiometer retrieval and the sonde). Since the authors are examining 3-6km average mixing ratio, it would be most useful to know the error characteristics of this average.

**This comment has been partially addressed in regards to the quality of the radiometer observations with the inclusion of several more references. Direct comparisons between the radiometer and the nearby radiosonde are given as Figs 1-4 but the error statistics are not. Because this study examines the changes in time, the relative quality of the radiometer observations as compared to the radiosondes is not particularly useful information for this study. If the temporal frequency of the radiosondes were higher, I would agree that an error evaluation may be helpful.**

Minor commments

1) Section 2, on the radiometer: Although the WV retrievals are performed on a 0.25km grid in the vertical, as the comparisons to radisonde profiles illustrate, the true vertical resolution is likely to be much lower. It would be useful to give this true resolution (noting that a neural network retrieval cannot supply this information, but there must have been studies using physical inversion techniques applied to similar radiometers that can supply this information).

**This is a very good point and has now been addressed with the inclusion of several more references.**

2) Is anything known about the radiative impact of smoke aerosol at frequencies used by the microwave radiometer? Presumably it is minimal, but it would be good to see some physical confirmation of this.

**This is an interesting question and one that did come to mind when I was first writing the paper but I could not find information related to the radiative impacts of the aerosols themselves with the frequencies used by the MWR. However, those frequencies are used because they do respond well to water vapor.**

3) Figure 6: Some explanation of the meaning and units of the colour scale needs to be given here. In particular the significance of the grey areas is not clear.

**This has been addressed.**

4) Figure 11: It is impossible to distinguish the aerosol zone from the ambient air, specially in panels b-d. Some adjustments may need to be made (e.g. to the colour scale?) on these figures.

**Thank you for pointing this out. The color was enhanced on this figure to make the smoke more obvious in the panels and I hope that it is now more easily seen.**

5) Figure 14, caption: Are panels (a) and (b) really both 1345 UTC?

**This was an error and has been fixed.**